# High-Accuracy Relative Pose Measurement of Noncooperative Objects Based on Double-Constrained Intersurface Mutual Projections

**DOI:** 10.3390/s22239029

**Published:** 2022-11-22

**Authors:** Yu Gan, Guangmin Li, Guodong Liu, Binghui Lu

**Affiliations:** School of Instrument Science and Engineering, Harbin Institute of Technology, Harbin 150001, China

**Keywords:** noncooperative objects, relative pose measurement, point cloud registration

## Abstract

Relative pose measurement for noncooperative objects is an important part of 3D shape recognition and motion tracking. The methods based on scanning point clouds have better environmental adaptability and stability than image-based methods. However, the discrete points obtained from a continuous surface are sparse, which leads to point-to-point dislocations in the overlapping area and seriously reduces the accuracy. Therefore, this paper proposed a relative-pose-measurement algorithm based on double-constrained intersurface mutual projections. First, the initial corresponding set was constructed using mutual projections between the areas with similar feature descriptors, and then the final corresponding set was determined through the rigid-transformation-consistency constraint to improve the accuracy of the matchings and achieve a high-accuracy relative pose measurement. In the Stanford dataset, the rotation error and translation error were reduced by 19.3% and 13.4%, respectively. Furthermore, based on the proposed evaluation method, which separated the error of the pose-measurement algorithm from that of the instrument, the experiments were carried out with a self-made swept-frequency interferometer. The rotation error was reduced by 39.8%, and the surface deviation was reduced by 4.9%, which further proved the advancement of the method.

## 1. Introduction

With the development of sensor technology, in application fields such as space on-orbit service, intelligent robots, automatic driving, UAVs, etc., it is challenging and important to complete object detection and use the captured information to calculate the position and attitude parameters. The pose-measurement methods for noncooperative objects based on the image method [1,2] can solve the 3D pose by extracting matching points from 2D images. It has the characteristics of good flexibility and low cost, but it is difficult to maintain the stability of the measurement results under the influence of interference factors such as light changes and a weak texture structure. On the other hand, with the continuous development of noncontact 3D measurement equipment such as LIDAR and laser scanners, point-cloud measurements are becoming more accurate and the amount of data is increasing. How to use the point cloud information obtained by the existing 3D scanning system to design a more robust and accurate pose measurement algorithm has become an important part of high-accuracy pose measurement for noncooperative objects. In the actual pose measurement process, it is usually necessary to register the partial overlapping point cloud data in different states. The relative pose-measurement method based on point clouds mainly includes auxiliary methods applied to cooperative objects and automatic methods applied to noncooperative objects. Auxiliary methods usually use artificial markers [3], rotating platforms [4], attitude sensors [5], and other auxiliary equipment to register the point clouds. However, the auxiliary measurement method has great limitations. Some measuring objects are not allowed to paste marker points, and the measurement efficiency is also greatly affected. The instruments must be calibrated accurately using the rotating platform and the attitude sensor, and extra inaccuracies will be introduced by the auxiliary devices used in the calibration. In noncooperative posture measurement, the automatic technique has gained increasing attention, which has the advantages of requiring no manual prior knowledge and high efficiency, extracting the similar structural features of the point cloud overlapping area, and solving the coordinate transformation matrix.

For the automatic measurement of the relative pose for noncooperative point clouds, Besl et al. [6] proposed an iterative closest point (ICP) algorithm based on point-to-point registration, which constructs the corresponding relationship through the nearest neighbor search, and then singular value decomposition (SVD) is used to minimize the Euclidean distance between the corresponding pairs to achieve the solution of the relative pose matrix. Chetverikov [7] improved the construction process of matching point pairs in the traditional ICP algorithm and used the proportion of overlapping regions of adjacent frame point clouds as prior information to filter corresponding pairs to achieve a better registration effect. He [8] et al. proposed a noncooperative pose-tracking method based on the geometric characteristics of point clouds, which improved the accuracy of matchings according to the density, normal vector, and other characteristics of the point clouds. Chen [9] et al. recognized the plane features on noncooperative objects and solved the relative pose of the feature points using the ICP algorithm. The plane feature extraction operation improved the accuracy and efficiency of the pose measurement. Zhu [10] et al. used a combination of local and global constraints to remove outliers in the initial matchings, which improved the accuracy of the matching pairs. Li [11] et al. proposed a rough registration algorithm based on principal component analysis combined with bounding boxes and FPFH descriptors, which overcame the problem that the ICP algorithm had high requirements for the initial values. Recently, He [12] et al. proposed a pose-measurement method based on geometric characteristics and used a random-sampling persistence algorithm to eliminate abnormal matching pairs, which improved the robustness of the relative pose measurement.

However, in the actual measurement process, the point cloud obtained by the discrete sampling of the laser scanning system is sparse, and the final matching points of the point-to-point pose measurement algorithm cannot one-by-one correspond to the actual measured surface, which results in the existence of pose-measurement errors. Low [13] investigated how to reduce the total distances between points and a tangent plane using the ICP approach. The object function included the normal vector, and the normal vector angle and the Euclidean distance were both minimized via least-squares optimization. Generalized ICP [14] was proposed to take into account the normal vector information. The influence of the sparsity on the discrete point cloud was somewhat reduced by this method, but it was not appropriate for complicated surface structures. On this basis, Jacopo [15] proposed the Normal ICP method to filter the matching pairs through the similarity of local area curvature, and minimize the distance from the point to the plane and the angle between the normal vector when solving the transformation matrix. Pengyu [16] used CoBigICP to expand GICP. The geometric information of the local corresponding region was introduced into the optimization function, and the registration effect is better. Although the above methods reduce the influence of point cloud sparsity by constructing an objective function to minimize the distance from the point to the tangent plane, the registration effect for some objects with complex surface morphology is not good. In this regard, the 3D-NDT method proposed by Das [17] used the probability density function to model the distance between points and realized the soft matching of correspondence. Evangelidis [18,19] proposed a point-cloud pose-measurement method based on the Gaussian mixture model. The target point cloud was represented by the Gaussian mixture model (GMM), and the transformation matrix was solved step by step by the Expectation Maximization (EM) algorithm. Choi [20] introduced the color information of matching point pairs into the probability cost function and constructed the probability distribution of the corresponding local area for the adjacent point set. However, the measurement equipment was required to provide RGB color information of the point cloud. Liu Tong [21] built a 3D model by fitting the target point cloud with a B-spline surface and gradually optimized the distance from the point to the 3D model in the registration process to improve the registration accuracy. However, this method required the fitting of closed surfaces and the high integrity of the target point cloud, so it could not be applied to the relative pose measurement for noncooperative objects with large nonoverlapping areas. Huang [22] proposed a variant ICP method based on an adaptive moving least squares (MLS) surface. MLS surface fitting was carried out in the local area of sparse point clouds, and the distance between points was replaced by the distance from the point to the projection of the MLS surface in the minimization objective function. Moreover, Huang further improved the fitting ability of traditional MLS surfaces to complex surfaces through a heterogeneous Gaussian kernel function and improved the registration accuracy. However, this method could not determine whether the projection area was accurate during the construction of the corresponding pairs, which led to lower accuracy of the final matchings or even the inability to obtain the correct matching pairs, which affected the final relative pose measurement’s accuracy.

In summary, to improve the accuracy of pose measurement, many researchers have conducted much research, but most of them mainly focused on improving the accuracy of the point-to-point pose measurement. There are few studies on nonpoint-to-point high-accuracy pose measurement for sparse point clouds, and there are still some problems in the existing methods, which cannot meet the actual measurement requirements. To overcome the influence of point-cloud sparsity on pose-measurement accuracy, this paper proposes a relative pose-measurement algorithm based on double-constrained intersurface mutual projections. First, the scanning interferometer was used to obtain the high-accuracy point-cloud information for the noncooperative objects, and then the similarity of the feature descriptor in the high-dimensional feature space was used to determine the bidirectional projection areas from the surface to the surface. Then, the position of the projection point was solved iteratively along the direction of the weighted normal vector to obtain the correspondences. Finally, the more accurate matching pairs were selected by the consistency constraints of rigid body transformation, and the relative pose matrix was calculated to achieve high-accuracy measurement of the relative pose for noncooperative objects. The innovation points of this paper are summarized as follows:(1)A heterogeneous moving least squares surface (HMLS) was constructed in the local mutual-projection area of point clouds to generate more accurate matchings, which reduced the impact of point-cloud sparsity on point-cloud pose measurement.(2)The similarity constraints of feature descriptors and the consistency constraints of rigid body transformation were applied to the selection of matching point pairs, and the accuracy of pose measurement for pairwise point clouds was improved;(3)An evaluation method for separating the pose-measurement error from the instrument error was proposed. Based on the Stanford dataset validation experiment, the actual noncooperative pose measurement experiments were further carried out through the self-made 3D swept-frequency interferometer, which reflected the significant advantages of the proposed method.

The main structure of this paper is as follows. Section 2 first analyzes the problems of the traditional ICP algorithm and describes the details of the relative pose-measurement algorithm based on double-constrained intersurface mutual projections. Section 3 provides the experimental results and the analytical discussions thereof. Section 4 concludes the work of this paper.

## 2. Method

The traditional ICP pose-measurement algorithm defines the target point cloud $P\in R^{3\times M_{1}}$ as the reference frame, and its pose state is fixed. The source point cloud is defined as $Q\in R^{3\times M_{2}}$, and its rotation matrix $R\in R^{3\times 3}$ and translation matrix $t\in R^{3\times 1}$ relative to the reference frame are the transformation parameters to be solved. $M_{1}$ and $M_{2}$ are the number of target points and source points, respectively. The corresponding set ${\left\{\left(p_{i},q_{i}\right)\right\}}_{i=1}^{N}$ is determined according to the Euclidean-distance-nearest principle between points. N is the number of correspondences in the overlapping area, which can be calculated according to the overlap rate. Take the sum of the squares of the point distances as the optimization function:(1)$$R^{*},t^{*}=\underset{R,t}{\mathrm{argmin}}\frac{1}{N}\sum_{i=1}^{N}{\left\|p_{i}-\left(Rq_{i}+t\right)\right\|}^{2}$$

$R^{*}$ and $t^{*}$ represent the rotation matrix and translation matrix of the source point cloud relative to the target point cloud, respectively. The point-to-point registration will seriously affect the accuracy of the relative position and attitude measurement due to the problem of dislocation for point clouds under different scanning angles. In contrast, the surface-to-surface registration method can fit the local shape of dense objects and make full use of the current point cloud structure information to improve registration accuracy. The main steps in proposing the method are as follows.

First, the point cloud of the noncooperative object was obtained by a 3D swept frequency interferometer. Then, after calculating the feature descriptor and normal vector of the point cloud, the multiscale feature descriptor was used to construct the initial corresponding set, and the mutual-projection area was determined according to the corresponding relationships. Local surface fitting was carried out for the point clouds, and the original points were mutually projected to establish a new corresponding set. Finally, the rigid-transformation consistent-distance constraint was used to filter the sets to obtain the final matchings and solve the coordinate-transformation matrix.

### 2.1. Principle of Point-Cloud Measurement for Noncooperative Objects

The measurement principle of the point cloud for noncooperative object is shown in Figure 1. The infrared laser emitted by the swept-frequency-interferometry ranging module was incident on the reflector. With the point of incidence as the origin, the coordinate system of the measurement system was established. We set the distance from the origin to the measured target as $R_{m}$. The scanning angles of the 2D turntable were the azimuth β and pitch α. In the scanning process, the 2D turntable read the values of the azimuth and pitch angles in real time and recorded the distance from the origin to the measured point measured by the swept-frequency-interferometry ranging module. Finally, Equation (2) was used to convert the 3D point cloud of the object from the polar coordinate system to the Cartesian coordinate system. After the current viewing angle scanning was completed, the target cloud $P\in R^{3\times M_{1}}$ and the source cloud $Q\in R^{3\times M_{2}}$ were obtained with the changing pose of the measured objects. $M_{1}$ and $M_{2}$ were the number of points of the target point cloud and source point cloud, respectively.
(2)$$\left\{ \begin{array}{l} x=R_{m}\sin\alpha \cos\beta \\ y=R_{m}\sin\alpha \sin\beta \\ z=R_{m}\cos\alpha \end{array}\right.$$

### 2.2. Projection from a Point to the MLS Surface

Local surface fitting is an important step in projection tasks. Amenta [23] set e(y,a) as the energy function of the MLS surface. A point y outside the surface was projected along the normal vector a of the corresponding local area. We defined the set of projection points that minimized the energy function as the MLS surface. ${\left\{p^{k}\right\}}_{k=1}^{K}$ was set as the k-nearest neighbors of the source point cloud relative to the target point cloud; Figure 2 shows the constructed MLS surface. The expression of the isosurface S(y) was:(3)$$\begin{array}{cc} S\left(y\right) & =\min\;e\left(y,a\right)=\min\;e\left(q_{m}+t\cdot a,a\right)\\ & =\min\left({\left(\sum_{k=1}^{K}{\left(q_{m}+t\cdot a-p^{k}\right)}^{T}a\right)}^{2}\theta \left(\left\|q_{m}+t\cdot a-p^{k}\right\|\right)\right) \end{array}$$

The initial point $q_{0}$ moved to the projection point $y=q=q_{m}+t\cdot a$ on the corresponding surface by minimizing the energy function e(y,a) along the weighted normal vector $n\left(q_{0}\right)$ of the K-neighborhood point set; the scalar t represented the single approximation step size. When  dS(y)/dt was zero, the energy function e(y,a) reached the minimum value. Then, the weighted normal vector $a=n\left(q_{m}\right)$ was introduced to transform the energy function into a function with a single-variable parameter. After that, Brent’s [24] algorithm was used to solve for the value of t in the nonlinear minimization function $e\left(t,n\left(q_{m}\right)\right)$. Finally, the position of the projection point $q=q_{m}+t\cdot n\left(q_{m}\right)$ was updated. When the difference between two adjacent projection points was less than the threshold $\delta =\left\|q_{m+1}-q_{m}\right\|$, the iteration was terminated. For some objects with smooth surfaces, we usually set parameter δ to $10^{-6}$.

The first term of the summation equation in the energy function represented the normal distance between the current iteration point $q_{m+1}$ and the nearest neighbor point $p^{k}$, and θ(⋅) represented the weighting function. a was the projection direction represented by the weighted normal vectors (3) of K adjacent points in the local surface region during each iteration.
(4)$$a=n\left(q_{m}\right)=\frac{\sum_{k=1}^{K}n_{pk}\theta \left(\left\|q_{m}-p^{k}\right\|\right)}{\left\|\sum_{k=1}^{K}n_{pk}\theta \left(\left\|q_{m}-p^{k}\right\|\right)\right\|}$$

The Gaussian kernel function θ(⋅) in the above MLS surface projection process has two different expressions. The Gaussian kernel function used in the traditional MLS surface is isotropic and can only describe the local surface with a single feature. However, the surface topography of real objects has anisotropic characteristics. When the flatness and steepness of the local point cloud in different directions are different, the traditional MLS surface projection results will have a large deviation. Therefore, the principal curvature parameters $k_{1}$ and $k_{2}$ in two orthogonal directions were introduced to construct the heterogeneous Gaussian kernel function. The heterogeneous Gaussian kernel function was defined as Equation (6), and the function widths could be expressed as $h_{1}=1/(\beta k_{1})$ and $h_{2}=1/(\beta k_{2})$. β was a scale factor that could indirectly control the width of the kernel function and affect the projection accuracy. The next section will discuss the influence of this parameter through experiments. $v_{1}$ and $v_{2}$ were the principal directions of curvatures *k*_1_ and *k*_2_, respectively.
(5)$$\theta \left(q_{m+1}-p^{k}\right)=\exp\left(-{\left\|q_{m+1}-p^{k}\right\|}^{2}/h^{2}\right)$$
(6)$$\theta^{\prime }\left(q_{m+1}-p^{k}\right)=\exp\left(-\beta^{2}\left(k_{1}^{2}{\left[\left(q_{m+1}-p^{k}\right)\cdot v_{1}\right]}^{2}+k_{2}^{2}{\left[\left(q_{m+1}-p^{k}\right)\cdot v_{2}\right]}^{2}\right)\right)$$

### 2.3. Local Feature Similarity Constraint

The above point-to-HMLS surface projection steps could generate new corresponding points on the local surface, but there was no corresponding determination mechanism to constrain the construction of the corresponding projection area. Inaccurate projection areas are prone to lead to inaccurate location information of projection points. Therefore, the feature similarity constraint was used to construct the intersurface mutual-projection regions. First, the multiscale feature descriptors $D_{P}\in R^{9\times M_{1}}$ and $D_{Q}\in R^{9\times M_{1}}$ of the target point cloud P and source point cloud Q were calculated as in reference [25]. This feature descriptor first determined the neighborhood point sets under different scale radii and then performed a principal component analysis on the point sets. The eigenvalues of different neighborhood point sets after the principal component analysis were normalized to obtain feature descriptors with a high robustness, high scale invariance, and high rotation invariance.

As shown in Figure 3, in the overlapping region of the target cloud and the source cloud, the MLS surfaces S(q) and S(p) were used to fit the local neighborhood. $q_{j}$ represented a point in the source cloud, and the KD-tree algorithm was used to build the nearest-neighbor index of the target cloud to obtain the initial correspondence $\left(q_{j},p_{j}\right)$, as shown in the matching point pairs connected by the dotted lines in the figure. Then, the local projection area of $q_{j}$ relative to the target cloud was fitted by ${\left\{p_{j}^{k}\right\}}_{k=1}^{K}$.

However, the projection region correspondence constructed by the single Euclidean distance nearest condition was not reliable. Therefore, according to the local feature similarity constraint, the correct corresponding point $p_{i}$ of $q_{j}$ was supposed to be in the neighborhood ${\left\{p_{j}^{k}\right\}}_{k=1}^{K}$. We set the neighborhood point set of the projection area corresponding to point $q_{j}$ as ${\left\{p_{j}^{k}\right\}}_{k=1}^{K}$ and the corresponding feature descriptor as $D_{p_{j}}$. Then ${\left\{D_{p_{j}^{k}}\right\}}_{k=1}^{K}$ represented the feature descriptor set of the neighborhood points. The neighborhood point $p_{i}$ that was nearest to the Euclidean distance of $D_{p_{j}}$ in the feature space was taken as the new corresponding point of $q_{j}$ to build a new correspondence $\left(q_{j},p_{i}\right)$, as shown by the solid line in Figure 3. To reduce the influence of outliers and noise points caused by measurement errors on the relative pose measurement, threshold $\varepsilon_{1}$ was set to filter the minimum Euclidean distance in the feature space, as shown in Equation (6). When the minimum value was less than $\varepsilon_{1}$, the relationship was considered a correct matching, and bidirectional projection areas $\left(p_{i},{\left\{q_{j}^{k}\right\}}_{k=1}^{K}\right)$ and $\left(q_{j},{\left\{p_{i}^{k}\right\}}_{k=1}^{K}\right)$ were obtained. Otherwise, the corresponding points were removed from the initial set. The selection of threshold $\varepsilon_{1}$ should not only filter outliers and noise points, but also should ensure the number of final matching pairs. The threshold $\varepsilon_{1}$ should be selected within the appropriate range. According to [25] and the previous test, we set $\varepsilon_{1}$ to 0.01.
(7)$$\left\|D_{q_{j}}-D_{p_{i}}\right\|<\varepsilon_{1}$$

### 2.4. Distance Constraint of an Intersurface Mutual Projection

The accuracy of matching pairs seriously affects the final relative pose-measurement results. Therefore, after the mutual projection of points to local surfaces, it was necessary to further use the rigid-transformation-consistency distance constraint to further filter the constructed intersurface mutual-projection point pairs.

As shown in Figure 4, the corresponding point pair $\left(p_{i},q_{j}\right)$ was obtained by the similarity constraints of the feature descriptors in the target cloud P and the source cloud Q. Then, the normal vector information and Equation (2) were used to project the two points to the corresponding regions ${\left\{q_{j}^{k}\right\}}_{k=1}^{K}$ and ${\left\{p_{i}^{k}\right\}}_{k=1}^{K}$, respectively, and obtain two pairs of new corresponding points $\left(p_{i},{p_{i}}^{\prime }\right)$ and $\left(q_{j},{q_{j}}^{\prime }\right)$. If the relationships $\left(p_{i},{p_{i}}^{\prime }\right)$ and $\left(q_{j},{q_{j}}^{\prime }\right)$ were correct, they should have satisfied the constraint of rigid-transformation consistency $\left\|p_{i}-{q_{j}}^{\prime }\right\|=\left\|q_{j}-{p_{i}}^{\prime }\right\|$. In the actual pose-measurement process, the condition that the distance of corresponding point pairs be completely equal was too strict. Therefore, the constraint condition was changed as follows:(8)$$\left|\frac{\left\|p_{i}-{q_{j}}^{\prime }\right\|-\left\|q_{j}-{p_{i}}^{\prime }\right\|}{\left\|p_{i}-{q_{j}}^{\prime }\right\|+\left\|q_{j}-{p_{i}}^{\prime }\right\|}\right|\leq \varepsilon_{2}$$

If the relative difference of the matching was less than the threshold $\varepsilon_{2}$, the relationships were regarded as the correct correspondences and added to the final matching point set. Otherwise, the corresponding relation was ignored. We set $\varepsilon_{2}$ to 0.005 according to the same principle as $\varepsilon_{1}$ in Section 2.3.

### 2.5. Relative Pose Measurement Based on Double-Constrained Intersurface Mutual Projections

To improve the accuracy of the relative-pose-measurement algorithm for noncooperative objects, the specific steps of the surface-to-surface registration algorithm based on double constrained intersurface mutual projections were as follows:

1. The swept-frequency interferometer was used to measure the noncooperative object, and the point cloud information was obtained according to Equation (2);

2. We preprocessed the input target cloud $P\in R^{3\times M_{1}}$ and source cloud $Q\in R^{3\times M_{2}}$. According to [25], we calculated the multiscale feature descriptors $D_{P}$ and $D_{Q}$, principal curvatures $k_{P}$ and $k_{Q}$, and normal vectors $n_{P}$ and $n_{Q}$;

3. The point number N in the overlapping area was determined according to the overlap rate parameter of the adjacent frames. The initial set of corresponding points ${\left\{{\left(p_{i},q_{i}\right)}^{l}\right\}}_{i=1}^{N}$ was constructed by the k-nearest neighbor search. Point $q_{i}$ in the source point cloud corresponded to the neighborhood point set ${\left\{q_{i}^{k}\right\}}_{k=1}^{K}$ of the target point cloud. According to the similarity of the feature descriptor and Equation (7), the corresponding pair set ${\left\{{\left(p_{i},q_{j}\right)}^{l}\right\}}_{i=j=1}^{N}$ under the similarity constraint of the feature descriptor was obtained. Thus, the bidirectional projection regions ${\left\{{\left(p_{i},{\left\{q_{j}^{k}\right\}}_{k=1}^{K}\right)}^{l}\right\}}_{i=1}^{N}$ and ${\left\{{\left(q_{j},{\left\{p_{i}^{k}\right\}}_{k=1}^{K}\right)}^{l}\right\}}_{j=1}^{N}$ were constructed.

4. We used Equation (3) to realize the mutual projection between each point and the corresponding area. In the projection process, the heterogeneous Gaussian kernel (6) was used to approach the step size step by step. Then, we constructed new corresponding sets ${\left\{{\left(p_{i},{p_{i}}^{\prime }\right)}^{l}\right\}}_{i=1}^{N}$ and ${\left\{{\left(q_{j},{q_{j}}^{\prime }\right)}^{l}\right\}}_{j=1}^{N}$;

5. We filtered the corresponding set with the rigid-transformation-consistent constraint (8) and added the correspondences $\left(p_{i},{p_{i}}^{\prime }\right)$ and $\left(q_{j},{q_{j}}^{\prime }\right)$ that satisfied the distance constraint to the final matching sets $P^{\prime }=\left\{p_{i},{q_{j}}^{\prime }|i=1,\ldots ,N,j=1,\ldots N\right\}$ and $Q^{\prime }=\left\{{p_{i}}^{\prime },q_{j}|i=1,\ldots ,N,j=1,\ldots N\right\}$;

6. We solved the coordinate-transformation matrix $R^{l+1}$ and $t^{l+1}$ based on Equation (1) and updated the source point cloud $Q^{l+1}=R^{l+1}Q^{l}+t^{l+1}$ iteratively until the convergence condition was satisfied.

The above algorithm looped through steps 2 to 5 until the difference in the transformation matrix was sufficiently small. When the number l of iterations exceeded the maximum number L, the pairwise registration process would end. L could prevent the algorithm from a failure to converge caused by a slight fluctuation in the registration results of the adjacent frames. The implementation details of the algorithm are shown in Algorithm 1.

**Algorithm 1:** Relative pose measurement based on double-constrained projection**Input:** Source point cloud **Q** and target point cloud **P** measured by the swept frequency interferometer**Output:** Transformation **R** and **t** that aligns **P** and **Q**1     Initial pose R0,t0 and maximum number of iterations *L*;2     Compute DP,DQ,kP,kQ,nP,nQ;3     Initial q = 0;4   **While** R,t are not converged or *l* < *L* do5        Initial set of corresponding points {(pi,qi)l}i=1N by kd-tree;6        **for** *i* = 1:N **do**7            **if** ‖Dqj−Dpi‖<ε18                  Build bidirectional projection region: {(pi,{qjk}k=1K)l}i=1N and {(qj,{pik}k=1K)l}j=1N;9            **end**10           Point to HMLS surface projection: {(pi,pi′)q}i=1N and {(qj,qj′)q}j=1N;11           **if** |‖pi−qj′‖−‖qj−pi′‖‖pi−qj′‖+‖qj−pi′‖|≤ε212                  Preserve the current corresponding point P′ and Q′;13         **else**14            Delete the current corresponding point;15         **end**16       **end**17       Compute Rq, tq through the final set of points P′ and Q′;18       Update the source point cloud: Ql+1=Rl+1Ql+tl+119       *l* = *l* + 1;20 **end**21 **return**
R, t;

## 3. Experiment and Analysis

### 3.1. Simulation of Relative Pose Measurement

To verify the advancement of the proposed pose-measurement algorithm based on double-constrained intersurface mutual projections, in the Stanford 3D scanning dataset (Available online: http://graphics.stanford.edu/data/3Dscanrep/ accessed on 10 October 2022), the first two perspectives on the Bunny, Dragon, Armadillo, and Happy Buddha models were used for the relative pose-measurement experiment. Each dataset contained partially overlapping point clouds and the true relative transformation of each frame. The accuracy of the pose measurement was measured via the rotation error $E_{R}$(°) and the translation error $E_{t}$ (mm):(9)$$E_{R}=\cos^{-1}\left(\frac{trace\left(RR_{GT}^{-1}\right)-1}{2}\right)\frac{180}{\pi },\;E_{t}={\left\|t-t_{GT}\right\|}^{2}$$

$R_{GT}$ and $t_{GT}$ were the respective real relative-rotation-transformation parameters and relative-translation-transformation parameters given in the Stanford dataset. R and t represented the relative-rotation-transformation parameters and relative-translation-transformation parameters solved by different algorithms. The overlapping area proportion of each pair of point clouds in the Bunny, Dragon, Armadillo, and Happy Buddha datasets were approximately 75%, 60%, 60%, and 60%, respectively; and the true relative rotation transformation angles were approximately 45°, 24°, 24°, and 24°, respectively. TriICP [6], point-to-plane ICP [13], GMM [19], CoBigICP [16], and HMLS_ICP [22] were used to conduct the relative pose-measurement experiment for noncooperative objects and compared it with the proposed algorithm. The Point-to-Plane ICP and TriICP algorithms were implemented through the open-source PCL library, which is based on the C++ development platform. The number of neighborhood points set in the Point-to-Plane ICP algorithm to solve the normal vector was 30. The overlap rate of point clouds with different viewing angles set in the TriICP algorithm was 60%. The effects of the GMM algorithm and CoBigICP algorithm were verified by the open-source code provided by Evalida [19] and Pengyu [16], respectively. The HMLS_ ICP algorithm mainly included two steps: solving the orthogonal curvature of the local point and projecting points onto HMLS surfaces. The reproduction process of this algorithm mainly referred to [22]. The proposed algorithm set the projection threshold, the similarity threshold of the feature descriptor $\varepsilon_{1}$, and the relative distance threshold of the consistency-constraint parameter of the rigid transformation $\varepsilon_{2}$ to 10^−6^ m, 0.01, and 0.005, respectively. The registration effect of the Bunny, Dragon, Armadillo, and Happy Buddha data is shown in Figure 5, in which the red point cloud represents the target point cloud, and the green point cloud represents the source point cloud.

First, the point clouds were roughly aligned using the algorithm proposed by Lei [25], and then different relative-pose-measurement algorithms were used for comparison. The initial values of the pose error and the pose-measurement error for different algorithms are shown in Table 1. The results with the lowest error are in bold. The table shows that the accuracies of the TriICP and Point-to-Plane ICP were better. The Point-to-Plane ICP algorithm realized point-to-plane registration by minimizing the distance between points in the normal vector direction. Although it could improve the relative rotation accuracy, the relative translation error increased, and the overall registration effect was worse than that of the TriICP algorithm. GMM replaced the Euclidean distance between points with a probability density function, so the correspondence between points was no longer absolute. Table 1 shows that the overlap rate of the point-cloud data had a great impact on the algorithm. CoBigICP also used local planes to represent point clouds, so the effect of pose measurement was poor when registering objects with complex surfaces. Although the point-cloud-registration method based on an HMLS surface obtained good pose-measurement results on objects with smooth surfaces such as Bunny, the pose-measurement error was large for objects with large surface undulations. The proposed algorithm filtered the projection area and matching point pairs using the feature-descriptor-similarity constraint and rigid-body-transformation consistency constraint, and it could improve the accuracy of the pose measurements on the basis of improving the accuracy of the matching pairs. The results showed that the rotation error of the proposed method was 19.6% lower than that of the traditional TriICP point-to-point pose-measurement algorithm while keeping the translation error unchanged. Compared with the existing optimal nonpoint-to-point method (HMLS-ICP), the average rotation error was reduced by 19.3%, and the translation error was reduced by 13.4%. However, the translation error of the proposed method in the Happy Buddha model was slightly larger than that of the other methods This was mainly because there were many areas with sharp surface fluctuations in the original Happy Buddha point clouds, which led to a decline in accuracy of the projection points in the process of manual projection for the proposed algorithm, thus affecting the pose-measurement results.

We used an Intel (R) I7 11800H 2.3 GHz CPU to measure the relative position and attitude of the Bunny dataset. The calculation times of the different algorithms were recorded as shown in Table 2. Compared with the TriICP, Point-to-Plane ICP, and CoBigICP algorithms, the proposed method had a higher measurement accuracy, but it required a greater number of nonlinear optimization operations, which resulted in an increased calculation time. This problem also existed for GMM and HMLS-ICP. The objective function of the GMM algorithm was the most complex, so the calculation time was longer than that of HMLS-ICP. Due to the interface mutual projection, the calculation time of the proposed algorithm was doubled compared with that of HMLS-ICP, but was roughly the same as that of the GMM algorithm. In future research work, we will further improve the operation efficiency of the proposed algorithm through algorithm optimization and hardware acceleration.

### 3.2. Experiment of Swept-Frequency Interferometry

Different from traditional laser-scanning equipment based on the time-of-flight (TOF) method, the 3D swept-frequency interferometer uses swept-frequency interferometry based on a frequency-modulated continuous wave (FMCW). By measuring the frequency difference between the transmitted light wave and the reflected echo of the object, the distance information between the measured target and the instrument can be calculated. This method has the characteristics of a large field of view and noncontact measurement; it is more suitable for the 3D pose measurement of noncooperative objects. To accurately evaluate the accuracy of the pose-measurement algorithm, avoid the influence of the measurement instrument’s own errors on the pose-measurement result, and separate the instrument error from the pose-measurement algorithm error, we proposed a method for pose-measurement evaluation. The self-made 3D swept-frequency interferometer was used to measure the relative pose of the Agriba gypsum model. Under the same visual angle, the red area and green areas of the gypsum model (as shown in Figure 6a) were scanned and measured twice in different steps, and all the point clouds were used as the original reference point clouds (as shown in Figure 6b). The red area was used as the target cloud, and the green area was used as the source cloud to perform the coordinate transformation, which was set manually. The relative pose was then calculated. The measured coordinate-transformation parameters were compared with the setting parameters, and the relative-pose-measurement accuracy of the dual-view point cloud was calculated. The transformed point cloud was compared with the reference point cloud after the same reconstruction, and the surface topography deviation was calculated.

The transformation results of 10° for the source cloud are shown in Figure 7a, in which it can be seen that the pose of the source cloud was shifted. After coarse registration, the transformation matrixes were solved using the TriICP, the HMLS-ICP, and the proposed algorithms. The relative pose-measurement results are shown in Figure 7b–d. Both the TrICP algorithm and the proposed algorithm could further improve the accuracy of the pose-measurement accuracy on the basis of coarse registration. In the pose-measurement experiments with different rotation angles, the registration results for the rotation-transformation parameters and translation-transformation parameters are shown in Table 3. In the actual process of the relative-pose-measurement experiment of the swept-frequency interferometer, the results of the position and attitude measurements using this algorithm and the HMLS-ICP algorithm were better than those of the TriICP algorithm. This was because the surface-to-surface registration algorithm could better overcome the sparsity of point clouds and improve the registration accuracy. Compared with HMLS-ICP, the rotation error of this algorithm was reduced by 39.8% while the translation was clearly increased, which showed that proposed algorithm had a higher advantage in the pose-measurement field that was more sensitive to rotation errors. In the following section, we will further verify the advancement of the algorithm through surface-deviation experiments. Furthermore, the registration results for different initial poses were almost the same, which verified the stability of the proposed algorithm.

### 3.3. Surface-Deviation Experiment

The relative pose of the source point cloud solved via point-cloud registration was used to transform its own coordinates, which allowed it to be registered to the target point cloud’s coordinate system so that the reconstructed point cloud of two frames could be obtained. Then, to further obtain the surface deviation in the pose-measurement results, the same surface-reconstruction algorithm was used for the reconstruction point cloud of different algorithms to obtain the triangular mesh representation. Then, we calculated the normal distance between a single triangular mesh patch and the neighboring points of the reference point cloud obtained in Section 3.2. The minimum normal distance was taken as the surface deviation value of the triangular patch. Finally, we calculated the surface deviation root-mean-square error (RMSE) value of the triangular mesh patches. Relative to the reference point cloud, the surface-deviation RMSE in the TriICP reconstructed cloud was 0.279 mm, and the maximum deviation was 1.155 mm. As shown in Figure 8a, there were large red areas with high deviations. The surface-deviation RMSE calculated using HMLS-ICP was 0.263 mm, and the maximum deviation was 1.182 mm. However, the surface-deviation RMSE of the proposed algorithm was 0.232 mm, and the maximum deviation was 1.139 mm. The RMSE value was 4.9% lower than that of HMLS-ICP. As shown in Figure 8c, the high deviation in the red area was less and was uniformly distributed, such as on the forehead and nose, which are marked with yellow boxes. Therefore, the experimental results verified the superiority of the proposed algorithm in improving the registration accuracy.

## 4. Conclusions

In the process of the relative pose measurement of noncooperative objects, image-based methods usually have certain limitations regarding the shape or texture of the object and are greatly affected by changes in the light intensity. In contrast, the method based on point clouds can better use the geometric information of objects and has better adaptability and stability. However, the sparsity of the discrete points measured by the 3D scanning system is inevitable. In this paper, to solve the problem that the existing point-to-point methods cannot meet the high-accuracy requirement due to this phenomenon, a relative pose-measurement method for noncooperative objects based on double-constrained intersurface mutual projections was proposed, which provided a new method for realizing surface-to-surface pose measurements. The method first searched for the bidirectional projection area through the similarity of multiscale feature descriptors, then performed point-to-surface projection and constructed a new corresponding set. Finally, the distance constraint of an intersurface mutual projection was used to filter the matching point pairs, which improved the accuracy of the final matchings and reduced the impact of point-cloud sampling sparsity on the pose measurements. Through the Stanford dataset and the self-made 3D swept-frequency interferometer, the relative-pose-measurement experiments were carried out, and the comparison results with the existing algorithms were given; these showed good advancement and practicality and verified that the proposed method had broad application prospects.

## Figures and Tables

**Figure 1 sensors-22-09029-f001:**
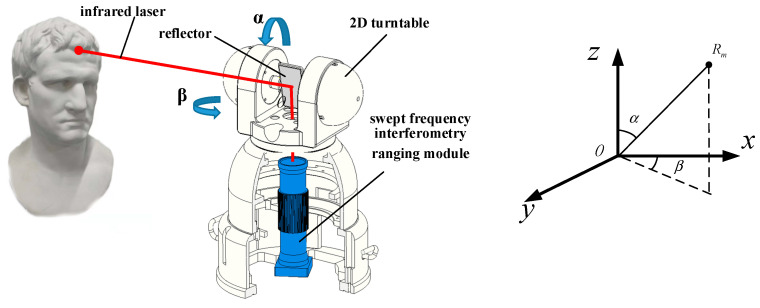
Schematic diagram of the point-cloud measurement principle for noncooperative objects.

**Figure 2 sensors-22-09029-f002:**
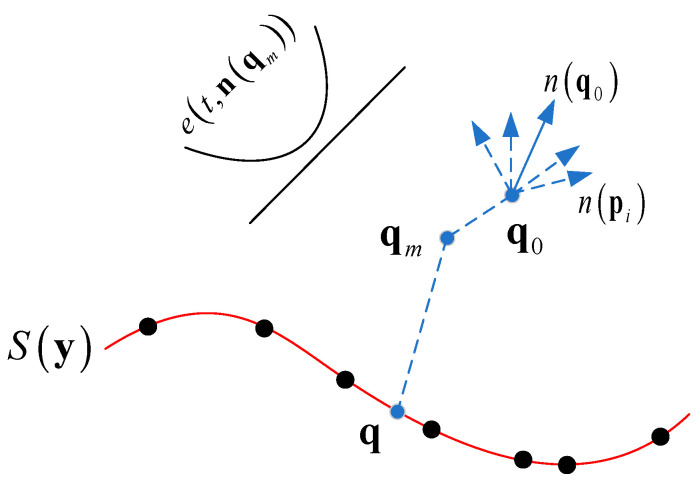
Projection of a point to the moving least squares (MLS) surface.

**Figure 3 sensors-22-09029-f003:**
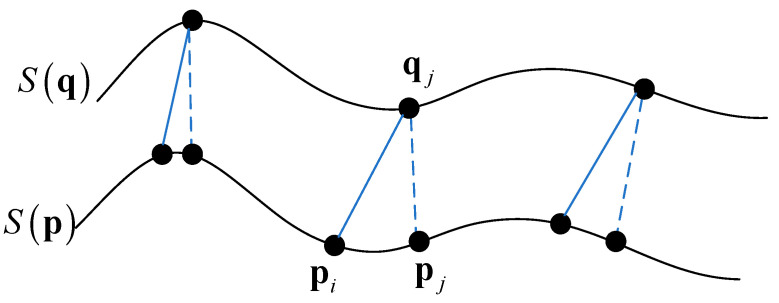
Local feature similarity constraint.

**Figure 4 sensors-22-09029-f004:**
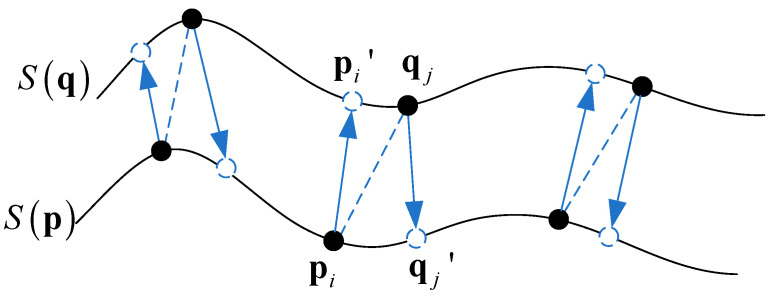
Distance constraint of intersurface mutual projection.

**Figure 5 sensors-22-09029-f005:**
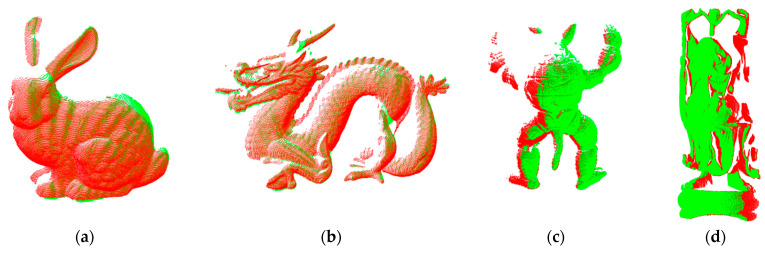
Original point-cloud data and registration rendering. The red point cloud represents the target point cloud, and the green point cloud represents the source point cloud. (**a**) Bunny; (**b**) Dragon; (**c**) Armadillo; (**d**) Happy Buddha.

**Figure 6 sensors-22-09029-f006:**
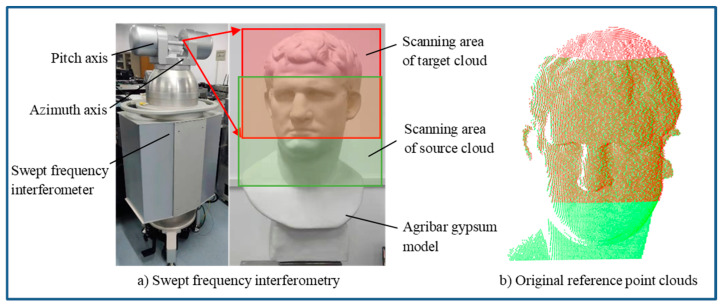
Schematic diagram of swept-frequency-interference pose-measurement experiment.

**Figure 7 sensors-22-09029-f007:**
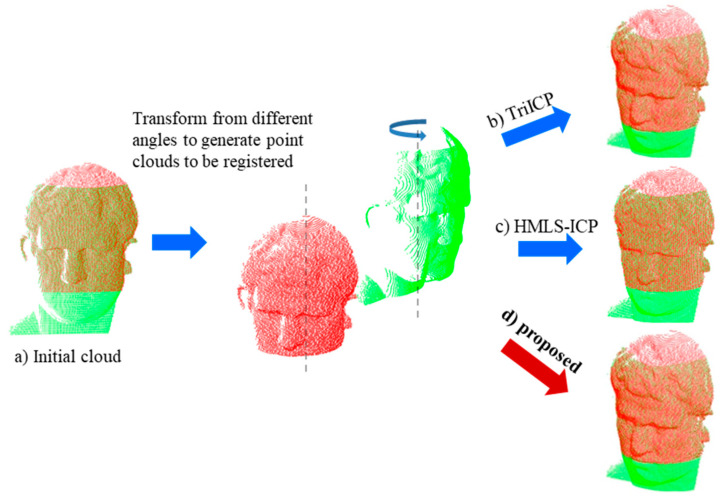
Results of the relative pose measurements using real point-cloud data.

**Figure 8 sensors-22-09029-f008:**
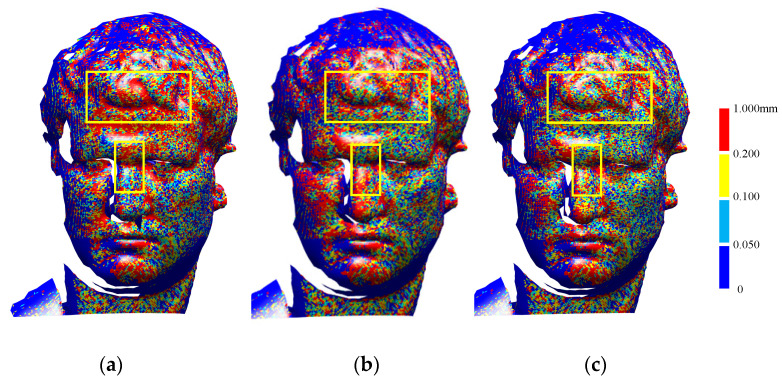
The deviations in surfaces reconstructed using different results of relative pose measurements of the reference cloud. (**a**) Surface deviation of TriICP; (**b**) surface deviation of HMLS-ICP; (**c**) surface deviation.

**Table 1 sensors-22-09029-t001:** Rotation errors and translation errors (°/mm) for different algorithms. (The results with the lowest error are in bold.)

	Bunny	Dragon	Armadillo	Happy Buddha	Average
Initial	0.8851/0.271	0.4973/0.256	1.2223/0.555	0.7526/0.783	0.8393/0.466
TriICP	0.1182/0.061	0.0524/0.129	0.0780/**0.090**	0.1869/0.355	0.1089/**0.159**
Point-to-Plane ICP	**0.0525**/0.186	0.0791/0.191	0.1180/0.157	0.1867/0.362	0.1091/0.224
GMM	0.1625/1.036	0.4528/0.324	0.3615/0.366	0.5484/**0.204**	0.3813/0.483
CoBigICP	0.0784/0.023	0.1152/0.267	0.1927/0.386	0.4669/0.301	0.2133/0.244
HMLS-ICP	0.0528/**0.018**	0.0771/0.189	0.1216/0.177	0.1826/0.364	0.1085/0.187
Proposed	0.0699/0.056	**0.0381/0.125**	**0.0615**/0.102	**0.1810**/0.366	**0.0876**/0.162

**Table 2 sensors-22-09029-t002:** Registration time (s) of different algorithms.

TriICP	Point-to-Plane ICP	GMM	CoBigICP	HMLS-ICP	Proposed
1.620	0.2336	605.9	1.680	360.3	672.7

**Table 3 sensors-22-09029-t003:** Rotation errors and translation errors (°/mm) under different transformation angles. (The results with the lowest error are in bold.)

	5°	10°	15°	Average
Initial	0.2696/0.445	0.3779/0.394	0.2935/0.581	0.3277/0.710
TriICP	0.1118/**0.190**	0.1289/**0.197**	0.1118/**0.198**	0.1175/**0.195**
HMLS-ICP	0.0597/0.197	0.0556/0.195	0.0586/0.205	0.0579/0.199
Proposed	**0.0362**/0.207	**0.0351**/0.207	**0.0331**/0.216	**0.0348**/0.210

## Data Availability

Not applicable.

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
