# Peer review of "High-Accuracy Relative Pose Measurement of Noncooperative Objects Based on Double-Constrained Intersurface Mutual Projections"

_sensors, 2022, doi:10.3390/s22239029_

Round 1
Reviewer 1 Report
This paper has certain innovation in theory and method, and has carried on the complete theory elaboration. Also, it has verified the effect of the proposed method with complete experiments. As a whole, it is worthy of recognition, while the following issues deserve attention:
1)For the sake of preciseness, it should be indicated that what the red and green point clouds in Figure 5 represent respectively, and the meaning of bold numbers in Table 1 should be supplemented.
2)It is suggested to frame the areas with obvious difference in effect (such as forehead and nose) in Figure 8.
3)In Section 3.1, data in Table 1 show that the translation error of the proposed method in the Armadillo model experiment is slightly larger than that of other methods. Please explain the difference in the effect of this model.
Reviewer 2 Report
The authors proposed the use of intersurface mutual projection to increase the accuracy of the point cloud matching. The quality of the manuscript is high and there are only few comments I would like to make.
1. On Page 5, some font adjustment might be needed. Its current state is a little messy.
2. On Line 217, what does "as usual" mean in this context?
3. On Line 345, the authors stated three sets of point cloud as Bunny, Dragon, and Happy Buddha. However, on Line 387, the last one is Armadillo.
4. On Line 359, there are two "sets", which might be an error, and the following sentence is wordy. Please consider modifying it.
5. In Table1, the authors highlighted the average translation error of the proposed method even though it is not the best one. It can lead to misunderstanding although the difference is very small. Moreover, can these errors be directly calculated for average without weighing? Should not the set that has higher amount of point cloud has higher weight than the smaller one?
6. In Table 2, TriICP has better translation errors than most of HMLS-ICP's ones. Please verify if TriICP's translation errors should be in bold instead of HMLS-ICP's ones?
7. In Section 3.2, the authors described the increased translation error as "basically unchanged" (Line 429). However, 7% of changing is arguably not "basically unchanged". In other scope of work that require the correctness of translation, this error might lead to the infeasibility of this proposed technique.
8. The authors should also raise the limitation of this proposed procedure for the readers. For example, in Section 3.2, although the rotation error is decreased, the translation is clearly increased. It would be beneficial for the readers also if the authors can compare the computational time between these techniques because by doing mutual projection, the computational time might increase doubly from back-and-forth projection.
